# Community Structure and Diversity of Endophytic Fungi in Cultivated *Polygala crotalarioides* at Two Different Growth Stages Based on Culture-Independent and Culture-Based Methods

**DOI:** 10.3390/jof10030195

**Published:** 2024-03-04

**Authors:** Kaize Shen, Yu Xiong, Yanfang Liu, Xingwang Fan, Rui Zhu, Zumao Hu, Congying Li, Yan Hua

**Affiliations:** 1Key Laboratory for Forest Resources Conservation and Utilization in the Southwest Mountains of China, Ministry of Education, Southwest Forestry University, Kunming 650224, China; ztushenkz@163.com; 2Yunnan Key Laboratory of Gastrodia and Fungi Symbiotic Biology, Zhaotong University, Zhaotong 657000, China; 15187086705@163.com (Y.X.); yanfang18469269355@163.com (Y.L.); m18314396046@163.com (X.F.); 15687032495@163.com (R.Z.); 15012364113@163.com (Z.H.); 3Yunnan Engineering Research Center of Green Planting and Processing of Gastrodia, Zhaotong University, Zhaotong 657000, China; 4Yunnan Yihua Agricultural and Biological Co., Ltd., Lincang 677300, China; licongy1235@163.com

**Keywords:** *Polygala crotalarioides*, endophytic fungi, fungal diversity, community structure, culture-independent and -dependent methods, anti-acetylcholinesterase

## Abstract

*Polygala crotalarioides*, a perennial herbaceous plant found in southwest China, has the potential to be used in the treatment of Alzheimer’s disease. Endophytic fungi that reside within medicinal herbs play an important ecological role in their host plants and can serve as a valuable source for identifying active components. However, little is known about the diversity, and structure of endophytic fungi in *P. crotalarioides*. In this study, we investigated the community structure and diversity of endophytic fungi in the leaves, stems, and roots of *P. crotalarioides* at both 1- and 2-year-growth stages using a modern culture-independent method using both culture-independent (high-throughput sequencing, HTS) and culture-based methods. Using HTS, our results revealed that the richness and diversity of endophytic fungi in *P. crotalarioides* varied depending on the organs and growth stages. Specifically, stems and leaves exhibited significantly higher diversity compared to roots. Additionally, the highest diversity of endophytic fungi was observed in the stems of the 2-year-old plants. At the genus level, *Fusarium*, *Colletotrichum*, and *Phoma* were the most abundant endophytic fungi in 1-year-old samples, while *Cercospora*, *Apiotrichum*, and *Fusarium* were prevalent in 2-year-old samples. A total of 55 endophytic fungal strains belonging to two phyla and 24 genera were isolated from 150 plant tissue segments using culture-based methods. The anti-acetylcholinesterase activity of these isolates was evaluated in vitro and five of them, *Phialophora mustea* PCAM010, *Diaporthe nobilis* PCBM027, *Fusarium oxysporum* LP41, *F. oxysporum* SR60, and *Phoma herbarum* SM81, showed strong activity (>50% inhibition rate). These findings will serve as a theoretical basis and practical guide for comprehending the structural composition, biological diversity and bioactivity of endophytic fungi in *P. crotalarioides*.

## 1. Introduction

Endophytic fungi are microorganisms that colonize the interior of plant organs such as leaves, stems, seeds, trunk, roots, fruits, and flowers in intracellular and/or extracellular spaces without causing symptoms of disease in host plants during part of or their entire life cycle [1,2,3]. High species diversity is a typical characteristic of endophytic fungi, as evidenced by the fact that it is quite common for endophyte surveys to find assemblages consisting of more than 30 fungal species per host plant species [4]. Plants with endophytic fungi are crucial reservoirs of fungal diversity and new taxa [5]. During plant growth, endophytic fungi display many substantial benefits, such as accelerating plant growth [6], enhancing stress resistance [7], promoting nutrient absorption [8], and resisting pathogens [9]. Furthermore, endophytic fungi are prospective producers of an abundant and dependable source of bioactive and chemically novel compounds with potential for exploitation in a variety of areas, such as medicine, agriculture, and industry [10]. In past decades, endophyte diversity investigations have used cultivation-dependent methodologies. However, the traditional culturing methods restrict the identification of endophytic fungi due to the challenges associated with isolating many of these fungi [11]. With the development of next-generation sequencing techniques, high-throughput sequencing (HTS) investigations of fungal communities were initiated [12]. In recent years, this method has been used to reveal the diversity of endophytic fungi in many medicinal plants, such as *Huperzia serrata* [13], *Alpinia zerumbet* [14], and *Sophora alopecuroides* [15]. Some studies showed that the HTS techniques reflect the diversity of endophytic fungi in hosts more comprehensively than traditional culture methods [11,16,17].

Medicinal plants are a reservoir of fungal endophytes, some of which can particularly produce compounds with the same or similar chemical structures and bioactivities to their host components [18]. The genus *Polygala* (Polygalaceae), with more than 600 species, is widely distributed throughout the world except in New Zealand, Polynesia, and Arctic zones [19]. Most of these species are commonly used as medicinal herbs by local people in some countries such as China, Korea, and Japan. There are about 50 species belonging to the genus distributed in China; among them, *Polygala crotalarioides* Buch. -Ham. Ex DC. is a perennial herbaceous plant widely distributed in southwest China [20]. Our previous work showed that xanthones and glycosides are the main secondary metabolites of *P. crotalarioides* [21]. Likewise other *Polygala* species, including *P. tenuifolia* [22], *P. japonica* [23], and *P. crotalarioides,* have potential therapeutic effects on neuroprotection [24]. In fact, it has been used locally as a folk tonic by the Yunnan Wa people in China to treat Alzheimer’s disease (AD). Medicinal plants have become an important resource for discovering endophytic fungi with bioactivity [25,26]. However, there is still a lack of knowledge about the endophytic fungi in *P. crotalarioides*.

AD is a progressively degenerative condition, often characterized by memory loss and behavioral changes, and it is predominantly observed among the elderly [27]. Presently, acetylcholinesterase (AChE) inhibitors that block the cholinergic degeneration of acetylcholine are endorsed as effective treatments for mild to moderate AD [28]. Pharmaceuticals such as huperzine A and galantamine, currently on the market, are identified as superior sources of acetylcholinesterase inhibitors [29,30]. More and more research has found that endophytic fungi are becoming a significant source of AChE inhibitors [31,32,33], and it is essential to pay more attention to the diversity and AChE-inhibitory activity of endophytic fungi in *P. crotalarioides*. Normally, *P. crotalarioides* must develop for at least two years before it may be utilized medically, because locals in Yunnan believe that two-year-old plants can accumulate enough medicinal ingredients to fully exert the therapeutic effect. This phenomenon stimulated us to wonder if there were changes in the structure and diversity of their endophytic fungi with growth time. Here, we assessed the composition and diversity of endophytic fungi from *P. crotalarioides* at 1- and 2-year-growth stages using both HTS and traditional culture-based methods. Furthermore, the AChE-inhibitory activity of isolated endophytic fungi was also evaluated to screen for active strains.

## 2. Materials and Methods

### 2.1. Collection and Processing of Samples

To compare the diversity of endophytes in different organs at two growth stages of *P. crotalarioides*, the 1- and 2-year-old plants were successively collected at the same site in mid-November 2021 and 2022. The plant sampling site was located in Shuangjiang County, Yunnan Province, P. R. China (23°28′18″ N, 99°49′54″ E), which is a main production area of *P. crotalarioides*. Three sampling points (3 m × 3 m) were set up using the diagonal sampling method, and the distance between each point was more than 150 m. Three biological replicates were collected in sterile plastic bags and brought into our laboratory within 24 h at about 4 °C using ice bags. The samples were then divided into leaves, roots, and stems with sterile scissors. Organs were cleaned under running water before being surface-disinfected using the following protocol: 75% ethanol for 1 min, 2% sodium hypochlorite for 3 min, 75% ethanol for 1 min, 5 rinses in sterile distilled water, and drying with sterile filter paper [34]. The leaves, stems, and roots of the 1-year-old samples were marked as PL1, PS1, and PR1. The 2-year-old samples were similarly labelled as PL2, PS2, and PR2. The last portion of the washing water (20 μL) was inoculated on PDA (potato dextrose agar) at 28 °C for 7 days to validate its disinfection effectiveness. The plant organs that had been surface-disinfected were put in sterile airtight plastic bags and chilled at −80 °C until DNA was extracted.

### 2.2. DNA Extraction, Amplification of ITS rDNA Region and High-Throughput Sequencing

Using the DN easy Plant Mini Kit (Qiagen, Valencia, CA, USA), all samples’ total DNA was extracted in accordance with the manufacturer’s instructions. The sample DNA concentration and quality were examined using 1% agarose gel electrophoresis and a NanoDrop One spectrophotometer (Thermofisher Scientific Inc., Waltham, MA, USA). Using 20–30 ng DNA of each sample as the template, the ITS1 variable region was amplified using universal primers ITS1F (5′-CTTGGTCATTTAGAGGAAGTAA-3′) and ITS2R (5′-GCTGCGTTCTTCATCGATG C-3′) [11]. The PCR assays were carried out in a final volume of 20 µL containing 2 µL of 10× Buffer, 10 ng of template DNA, 2 µL of dNTPs, and 0.8 µL of each primer, 0.2 µL of BSA, and ddH_2_O. The ABI GeneAmp^®^ 9700 (Applied Biosystems, Foster, CA, USA) apparatus was used for amplification, and the following parameters were set: initial denaturation at 95 °C for 3 min, 35 cycles of denaturation at 95 °C for 30 s, annealing at 55 °C for 30 s, elongation at 72 °C for 45 s, and a final extension at 72 °C for 10 min. The PCR products were detected using electrophoresis on 2% agarose gel. Purified amplicons were pooled in equimolar and paired-end sequenced on an Illumina MiSeq PE300 platform (Illumina, San Diego, CA, USA) according to the standard protocols by Majorbio Bio-Pharm Technology Co., Ltd. (Shanghai, China). The data generated from the Illumina sequencing samples were deposited into the NCBI Sequence Read Archives (SRA) as Accession Number PRJNA973057.

### 2.3. Sequence Processing and Data Analysis for HTS

Each sample’s reads were spliced through overlap using Flash version 1.2.11, and the generated sequences were used as raw tags. Fastp version 0.19.6 was used to demultiplex and quality-filter the raw sequencing reads. With 97% pairwise identity, Uparse version 11 further grouped the final effective tags into operational taxonomic units (OTUs). All data are expressed as mean values, and one-way analysis of variance (ANOVA) was used to conduct the analysis. The representative sequences of OTUs were used to perform taxonomic analysis via the UNITE database using QIIME version 1.9.1Venn diagrams, microbial community barplots and pieplots, and rarefaction curves were generated using R software version 3.3.1. Using Mothur version 1.30.2, alpha diversity indices such Chao1, Ace, Shannon, and Simpson were assessed. Tukey’s HSD test was used to assess variances among means of these diversity indices at a significant level (*p* < 0.05). To analyze how closely various samples of species diversity resembled one another, beta diversity indices were assessed using the Qiime software version 1.9.1. The relationships between the fungal community structures in various samples were examined using Principal Coordinate Analyses (PCoA) using the Bray-Curtis distances. The above analysis was performed on the online platform of Majorbio Cloud Platform (www.majorbio.com) accessed on 10 August 2023.

### 2.4. Isolation and Identification of Endophytic Fungi with Culture-Based Method

After being surface-disinfected, plant samples were cut into small slices (0.2 to 0.5 cm) and put on PDA Petri dishes (10 cm diameter and each piece was spaced from 1.5 to 2 cm) with 100 µg/L ampicillin and streptomycin. The Petri dishes were parafilm-sealed and cultivated for 7–10 days at 28 °C. Fungal colonies were inoculated onto fresh PDA Petri dishes. The purified fungi were stored at −80 °C in a 50% glycerol solution at the College of Agronomy and Life Sciences of Zhaotong University.

Fungal genomic DNA was extracted using the CTAB method following the instructions on the Omega fungal genomic DNA extraction kit (Omega Bio-Tek, Norcross, GA, USA). A NanoDrop One/OneC microvolume UV-Vis spectrophotometer (Thermo Fisher Scientific, Waltham, MA, USA) was used to measure the DNA concentration (>100 ng/L in a volume of 25 µL). DNA amplification was performed on a volume of 60 μL which consisted of 42.6 μL sterile ddH_2_O, 6 μL buffer (10×), 1.2 μL 2 mmol dNTP, 1.5 μL 10 pmol of primers ITS1 (5′-TCCGTAGGTGAACCTGCGG-3′) and ITS4 (5′-TCCTCCGCTTATTG ATATGC-3′) [35], 1.2 μL 5U Taq DNA polymerase, and 6 μL DNA template. The PCR cycling protocol consisted of preheating at 94 °C for 5 min, 35 cycles of 94 °C for 1 min 30 s, 53 °C for 1 min 30 s, 72 °C for 2 min, and a final extension step at 72 °C for 10 min. Subsequently, 1% agarose gel electrophoresis was used to analyze 5 µL of PCR products. Purification and sequencing of PCR products were performed at Kunming Shuoqing Bio-Technique Co., Ltd. (Kunming, China). The amplified ITS sequences and existing species sequences in GenBank were analyzed using BLAST (http://blast.ncbi.nlm.nih.gov/Blast.cgi) accessed on 10 August 2023. The obtained ITS sequences were submitted to NCBI GenBank.

### 2.5. AChE-Inhibitory-Activity Screening of Isolated Endophytic Fungi

The individual isolated endophytic fungi were inoculated in PDB media at 180 rpm at 28 °C for 15 days in a rotary shaker. The fungal fermentation liquid was filtered through gauze, filter paper, and a 0.22 μm microporous filter membrane in succession to remove germs. For subsequent usage, the endophytic fungal fermentation liquid was gathered and kept in a freezer at 4 °C. The in vitro AChE-inhibiting activity of the fungi was determined with a modified method described by Kadiyala et al. [36]. In summary, a 96-well plate was prepared by adding 125 µL of 3 mM DTNB, 25 µL of 15 mM ATCI, 50 µL of phosphate buffer (pH 8; 50 mM), and 10 µL of the sample (fermentation liquid) to each well. Absorbance at 405 nm was recorded every 13 s for a total of 65 s to establish the baseline. The reaction was initiated by adding 0.20 U/mL of 25 µL AchE to each well. Absorbance at 405 nm was then monitored every 13 s for a total of 104 s. To eliminate any non-enzymatic changes in absorbance, the absorbance before and after the addition of the enzyme was subtracted. The percentage inhibition of AChE was calculated by comparing the rates of the samples to a control (50% hydroalcohol in phosphate buffer). PDB media was used as the negative control. Galanthamine (1 μg/mL) was used as the positive control. The percentage inhibition of AchE was determined using the formula: %I = (A[Control] − A[Sample]/A[Control]) × 100. Where A[Control] is the absorbance of the control reaction (except test extracts) and A[Sample] is the absorbance of sample reaction. The results of the experiment are presented as the mean ± SD (standard deviation) from three independent replicates.

## 3. Results

### 3.1. Surface-Disinfection Efficiency

After cultivation of the last portion of the washing water, the absence of fungal colonies on the PDA plates showed that surface-disinfection was successful.

### 3.2. Analysis of Endophytic Fungal Community Structures with HTS

#### 3.2.1. Overview of Sample Sequences

HTS was used to evaluate *P. crotalarioides* leaves, stems, and roots at two growth stages (three biological replicates for every organ sample). The average lengths of the sequences from the leaf, stem, and root of 1-year-old samples were 239, 234, and 260 bp, respectively; these lengths largely fell within the range of 220–280 bp. The number of effective tags per sample ranged from 127,966 to 181,913. The average length of the sequences from the three organs in 2-year-old samples was 232, 252, 226 bp, and the range of the effective tags was 205,448 to 229,716. The sequencing depth was appropriate because the Sobs index on the OTU level tended to plateau as illustrated in Figure 1. Meanwhile, the majority of the fungal diversity in the samples used in this analysis appears to be adequately represented by the ITS libraries, as shown with coverages of more than 0.999 following chimeric particle removal and filtering (Table 1). This meant that the structures of the endophytic fungal samples were accurately reflected in the sequencing results.

#### 3.2.2. Analysis of Operational Taxonomic Units (OTUs)

After quality filtering, a total of 1,114,159 high-quality sequences with an average sequence length of 240 bp were extracted from the samples. The OTU abundance was normalized using the sample containing the least sequences (31, 120). At a 97% similarity level, 1190 OTUs were obtained, removing the ones matching to the sequence of chloroplasts and mitochondria (Appendix A). Some Venn diagrams (Figure 2) were constructed to depict similar and overlapping OTUs among all samples. The unique OTUs among the six groups collected at different stages were 77, 46, 63, 63, 597, and 60, respectively, as presented in Figure 2a. Obviously, only eight fungal OTUs were common among all six groups, which accounted for 0.67% of the total number of OTUs. All of the OTUs were assigned into six phyla, 23 classes, 72 orders, 160 families, 290 genera, and 416 species excluding the unclassified taxa. As displayed in Figure 2b, in the 1-year-old samples, a total of 355 OTUs (Appendix A) were obtained with 204 in PL1, 164 in PS1, and 121 in PR1. In total, 54 OTUs were common between PL1 and PS1, 14 OTUs were common between PL1 and PR1, and 7 OTUs were common between PR1 and PS1. The number of unique OTUs of the three organs of the 1-year-old samples were 110, 76, and 69, respectively, which accounted for 30.9%, 21.41%, and 19.44% of the total OTUs. Simultaneously, only 34 endophytic fungal OTUs were common among different organs, which were 9.58% of the total number of OTUs. Figure 2c showed the Venn diagram of OTUs in the 2-year-old samples. A total of 981 OTUs (Appendix A), significantly more than the 1-year-old samples, were obtained from the 2-year-old samples, with 218, 829, and 143 OTUs in PL2, PS2, and PR2, respectively. There were 117, 4, and 44 OTUs common between PL2 and PS2, PL2 and PR2, PR2 and PS2, respectively. The unique OTUs in PL2, PS2, and PR2 were 75, 646, and 73, which were 7.65%, 65.85%, and 7.44% of the total OTUs, respectively. The most OTUs among the three organs were 22 in the 2-year-old samples. Compared with the 1-year-old samples, changes in the 2-year-old ones were observed, such that OTUs in the whole plant and different organs were significantly more, respectively, indicating significant variation in the fungal community structure.

#### 3.2.3. Alpha Diversity Analysis of the Endophytic Fungal Communities

Alpha diversity indices (Table 1) presented differences among all samples of *P. crotalarioides*. Using a higher Shannon index and lower Simpson index, we indicated the diversity of the fungal communities in the samples. The highest endophytic fungal community diversity was observed in PS2. In contrast, the lowest fungal community diversity was observed in PL2. Chao and Ace indices are used to evaluate the community richness. The highest endophytic fungal community abundance was in PS2, and the lowest fungal community abundance was in PR2. These findings indicated that the richness and diversity of the endophytic fungi of *P. crotalarioides* were quite different in different organs and at different growth stages. In the 1-year-old samples, the fungal communities’ richness and diversity was highest in the stem.

#### 3.2.4. Beta Diversity Analysis of the Endophytic Fungal Communities

Based on the ANOSIM inter-group difference test using the Bray–Curtis distance algorithm, a principal coordinates analysis (PCoA) at the OTU level of analysis was carried out to explore the beta-diversity of endophytic fungi. The first principal coordinate (PC1) and the second principal coordinate (PC2) showed a representative contribution rate of 22.12% and 19.79%, respectively, with a total explanation of 41.91%. As can be seen from Figure 3a, the overlap degree of fungal communities in the samples of the two growth stages was low, indicating different endophytic fungal community structures in the two different growth stages (R = 0.8724, *p* = 0.0010). Significant differences in the community structure of endophytic fungi were also observed in different organ groups except PL1 and PL2, regardless of whether the samples were annual or biennial. This result was confirmed using a hierarchical clustering tree in Figure 3b. For 1-year-old samples, PL1 and PS1 clustered together in one branch, while PR1 formed a separate branch. For 2-year-old samples, a PL2 group also formed a separate branch, and PS2 and PR2 merged together in one branch. Among samples of the two stages, PR1 and PR2 clustered together. This indicated that the endophytic fungal communities of the 1-year-old leaves and stems were similar to each other but distinct from the endophytic fungal communities in the 1-year-old roots.

#### 3.2.5. Composition of Endophytic Fungal Communities

When performing the taxonomic analysis, sequence reads were normalized at 31,120 minimum number of sequence reads per sample. In total, all sequences were assigned into six phyla, 23 classes, 72 orders, 160 families, 290 genera, and 416 species excluding the unclassified taxa. At the phylum level, Ascomycota was dominant in all of the samples with mean relative abundances of 85.60%, 85.55%, and 66.22% in leaves, stems, and roots of the 1-year-old samples, and 98.43%, 90.76%, and 55.16% in 2-year-old samples (Appendix A). As shown in Figure 4, at the same growth stage, there were significant differences in the endophytic fungus community composition and relative abundance among different organs. Similarly, community composition and relative abundance of endophytic fungi from the same organs changed with their stage of growth (Table 2).

At the genus level, excluding unclassified and relatively small (<0.01) fungal species (Appendix A), *Colletotrichum* (31.17%), *Phoma* (18.85%), and *Plectosphaerella* (4.16%) had the highest relative abundance in PL1. *Phoma* (17.75%), *Colletotrichum* (12.06%), and *Plectosphaerella* (7.56%) were the three most prevalent genera in PS1. The three most dominant genera in PR1 were *Fusarium* (44.24%), *Apiotrichum* (5.43%), and *Aspergillus* (2.06%). Twenty-nine genera were commonly found among three organs of 1-year-old samples, mainly *Fusarium* (19.14%), *Colletotrichum* (17.31%), *Phoma* (13.91%)*, Plectosphaerella* (4.47%)*, and Apiotrichum* (3.35%) (Table 2). In the 2-year-old samples, the composition of the endophytic fungi varied at the genus level. The main genera of PL2 were *Cercospora* (58.29%), *Calonectria* (9.77%), and *Colletotrichum* (8.64%). The dominant genera of PS2 were *Colletotrichum* (14.43%), *Exophiala* (8.90%), and *Cladophialophora* (7.75%). The main genera of PR2 were *Apiotrichum* (32.66%), *Fusarium* (19.09%), and *Exophiala* (13.82%). There were 27 common genera among the three organs from 2-year-old samples, with the preference of *Apiotrichum* (28.24%), *Fusarium* (21.56%), *Calonectria* (9.37%), *Papiliotrema* (7.37%), and *Acremonium* (6.13%) (Table 2).

At all taxonomic levels, there were more endophytic fungal species in 2-year-old stems and roots than those of 1 year old, but the opposite was true in leaves (Appendix A). At the genus level, PL1 had 99 genera, slightly more than PL2 with 92 genera. PS1 had 99 genera which was much fewer than PS2 with 276 genera. PR1 had 55 genera which was fewer than PR2 with 86 genera. There were 43, 65, and 24 common genera in leaves, stems, and roots between the two different-stage samples (Table 2). Between PL1 and PL2, the common genera mainly were *Cercospora* (33.69%), *Colletotrichum* (24.04%), *Phoma* (10.65%), *Phyllosticta* (4.90%), and *Diaporthe* (2.09%). Between PS1 and PS2, *Colletotrichum* (15.84%), *Phoma* 10.71%), *Fusarium* (6.82%), *Plectosphaerella* (5.13%), and *Cladophialophora* (4.64%) were dominant common genera. Between PR1 and PR2, *Fusarium* (43.62%), *Apiotrichum* (26.24%), *Papiliotrema* (3.12%), *Aspergillus* (1.61%), and *Cutaneotrichosporon* (1.60%) were dominant common genera.

### 3.3. Analysis of Culturable Endophytic Fungus Community Structure

In total, 55 strains of endophytic fungi were isolated from *P. crotalarioides* with culture-based methods for the first time (Table 3 and Appendix A). Colonies of some endophytic fungi isolated from *P. crotalarioides* on PDA cultured for 5 days are presented in Appendix A. There were 26 and 29 fungal strains isolated from 75 tissue segments of 1- and 2-year-old *P. crotalarioides* samples, respectively. The ITS rRNA gene sequences of these strains with accession numbers OR592306-OR592331 and OR597904-OR597932 were compared with the ones deposited in the GenBank database.

In the 1-year-old samples, the 26 strains were grouped into five classes [Dothideomycetes (23.08%), Eurotiomycetes (19.23%), Leotiomycetes (11.54%), Saccharomycetes (3.85%), and Sordariomycetes (38.46%)] within the phylum Ascomycota. At the genus level, there were 13 genera, with the most abundant endophytic fungi being *Fusarium*, *Phialophora*, *Phoma,* and *Plectosporium*. There were 10, 9, and 7 strains from leaves, roots, and stems, respectively. The common species were not found in three organ segments. The identified species were different in different organs except that *Coniochaeta velutina* and *Phialophora mustea* were found in both leaf and root segments.

The 29 isolates from the 2-year-old samples were categorized into five classes [Sordariomycetes (51.72%), Dothideomycetes (27.59%), Eurotiomycetes (10.34%), Agaricomycetes (6.90%), Pezizomycetes (3.45%)]. They were further classified into 15 genera, comprising 1 genus of Basidiomycota and 14 genera of Ascomycota. *Nigrospora*, *Colletotrichum*, and *Alternaria* were prevalent genera. Out of these strains, 14, 8, and 7 were from leaf, root and stem tissues, respectively. From the leaf and root samples, *Alternaria alternata* and *Colletotrichum horii* were both isolated. *Fusarium oxysporum* and *Nigrospora chinensis* were prevalent in leaf and stem samples. *Schizophyllum commune* was commonly found in the segments of roots and stems.

### 3.4. AChE-Inhibiting Activity of Isolated Endophytic Fungi

We evaluated the anti-AChE activity of the isolated endophytic fungi using their PDB fermentation broth (Table 3). Under the tested conditions, *Phialophora mustea* PCAM010 and *Diaporthe nobilis* PCBM027 from 1-year-old samples and *Fusarium oxysporum* LP41, *F. oxysporum* SR60, and *Phoma herbarum* SM81 from 2-year-old samples showed strong inhibitory activity with more than 50% inhibition. Sixteen strains of endophytic fungi showed some inhibitory effects, with inhibition rates ranging from 10% to 50%. The remaining 34 strains showed no inhibition of AChE (<10% of inhibition rate).

## 4. Discussion

Excavating potential endophytic fungi from more plants is crucial since mounting research indicates that these fungi are a significant source of a variety of compounds [37,38,39,40,41]. Since Stierle et al. [42] isolated a taxol-producing endophytic fungus from *Taxus brevifolia* in 1993, and the research on endophytic fungi in medicinal plants has become a hot topic for scientists worldwide. Medicinal plants—known for their therapeutic properties—often harbor a unique and diverse set of these fungi, which may be a source of novel bioactive compounds with potential medicinal applications [26]. The fungi found within these plants may produce similar or entirely new bioactive substances as part of their natural symbiotic relationship, which could lead to the development of new drugs and therapeutic agents [43]. The notion that endophytic fungi from medicinal plants are more likely to yield compounds with biological activity is supported by various studies. For instance, Strobel and Daisy highlighted in their seminal paper the vast potential of endophytes as producers of novel bioactive metabolites that could have applications in medicine, agriculture, and industry [44]. Additionally, Wang et al. demonstrated that endophytic fungi from medicinal plants may produce a plethora of novel compounds with antimicrobial, anticancer, and other health-related benefits [45]. Therefore, the focused exploration of endophytic fungi within medicinal plants not only helps in conserving these plants by uncovering alternative sources of therapeutic compounds but also propels the discovery of new drugs. In recent times, the increasing scarcity of medicinal plants has underlined the urgency of researching their associated endophytic fungi. Over decades, many active fungal endophytes which can produce various compounds, even the same secondary metabolites as their hosts, have been isolated and identified. A typical example is huperzine A produced by *Huperzia serrata*, an AChE inhibitor to treat early- to mid-stage Alzheimer’s disease. Various endophytic fungi isolated from *H. serrata* have provided a promising alternative source to produce Huperzine A [46], while more than 200 secondary metabolites have been characterized from endophytic fungi from this plant [47]. More and more research on traditional medicinal plants, such as *Panax notoginseng* [48], *Paeonia lactiflora* [49], *Glycyrrhiza uralensis* [50], *Sophora alopecuroides* [51], and *Vernonia anthelmintica* [52], have promoted the discovery of endophytic fungi producing bioactive compounds.

*Polygala* plants are rich sources of promising microbiomes [53]. An unidentified strain of *Colletotrichum* isolated from *Polygala elongata* was reported as a potential source of a natural antioxidant against ABTS and DPPH radicals [54]. *Burkholderia cepacia* COPS, a bacterial strain isolated from *Polygala paniculata* roots, had moderate antagonistic activity against *Acinetobacter baumannii* ATCC 19606 and *Escherichia coli* ATCC 25922 and potent cytotoxic effects on *Leishmania infantum* and *Leishmania major* promastigote [53]. These evidently manifest in plants of *Polygala* possibly being considered a rich source of microbes for the fight against some diseases. However, to our knowledge, there are still few reports on endophytic fungi in *Polygala* plants. In our study, HTS and culture methods were used to study the diversity of endophytic fungi in *P. crotalarioides* at 1- and 2-year-growth stage.

HTS is a tool to study endophytic fungi which may provide further opportunities to disclose unknown functions of endophytic communities [55]. Our study supplied enough insights to dig deeper into the endophytic resources of *P. crotalarioides* using HTS. Like many other reported plants, Ascomycota is the dominant phylum in either 1- or 2-year-old samples. Comparing the samples at the two growth stages, *Fusarium*, *Colletotrichum,* unclassified Ascomycota, *Phoma*, and *Apiotrichum* were the dominant genera in 1-year-old samples, while *Cercospora*, *Fusarium*, *Apiotrichum*, *Colletotrichum*, and *Exophiala* were dominant in 2-year-old samples at the genus level. To some extent, we have presented the comprehension of the fungal endophytes in this plant using HTS, along with the detection of various unclassified or unidentified fungi, implying that new microbial resources present in this medicinal plant.

It is worth discovering the endophytic fungal isolates from *P. crotalarioides* using culture-dependent methods. In the study, we isolated 55 strains of endophytic fungi cultivated on PDA medium, which was obviously much fewer than those determined using culture-independent approaches. HTS technology has the ability to detect a greater number of fungi compared to the culture-based methods when investigating fungal species diversity. In spite of this, using a combination of culture-based and culture-independent methods will increase the likelihood of accurately estimating the endophytic fungal community more than using either approach alone [56]. HTS provides a more comprehensive reference for identifying endophytic fungi. Our study, revealed that there were many unclassified or unidentified fungi determined using HTS. However, just some known strains were isolated using culture methods for the limitation of cultivation conditions. To ensure optimal isolation of endophytic fungi, it is essential to explore diverse cultivation conditions based on the specific characteristics of the target strains [57]. The key factors, such as media [58], inoculation temperature [59], and incubation period [60], should be considered for use in future studies for finding culturable endophytic fungi in *P. crotalarioides*.

A host’s age is one of the factors that affect the distribution and population structure of endophytes [61]. For example, fungal endophytes of *Panax ginseng* were distributed depending on the age of plants based on the research on 1-, 2-, 3-, and 4-year-old ginseng roots [62]. A study on the *Gentiana officinalis* also found that the endophytic communities can be influenced by plant age [63]. In our study, the community and diversity of endophytic fungi in *P. crotalarioides* were found to have notable differences between 1- and 2-year-old samples both using HTS and culture-based methods. Specifically, the 2-year-old samples exhibited a higher abundance of endophytic fungi at the order, family, genus, and species levels in all organs compared to the 1-year-old samples. These results further supported the notion that composition of the endophytic fungal communities depends on the plant age. *P. crotalarioides* must generally be grown for at least two years before being utilized medicinally in Yunnan, China. However, it is unknown whether the production and accumulation of active components in this plant is associated with endophytic fungal diversity.

Our study revealed that several endophytic fungal strains displayed significant inhibitory activity against AChE. Among them, *Phoma herbarum* SM81 demonstrated the highest inhibitory activity at 89.88% inhibition, followed by *Diaporthe nobilis* PCBM027 (65.92% inhibition), *Fusarium oxysporum* LP41 (57.88% inhibition), *F. oxysporum* SR60 (59.13% inhibition), and *Phialophora mustea* PCAM010 (54.02% inhibition). These findings are of great significance as AChE inhibitors are commonly used in the treatment of AD. AChE is responsible for the breakdown of acetylcholine, a neurotransmitter involved in cognitive function. Inhibiting this enzyme can increase acetylcholine levels and improve cognitive function in individuals with AD. Interestingly, previous studies have also reported the presence of AChE-inhibitory compounds in endophytic fungi [64]. This demonstrates the potential of endophytic fungi as a source of novel bioactive compounds for the development of new drugs. The identification of these endophytic fungal strains with strong inhibitory activity against AChE provides valuable insights into their potential as therapeutic agents for AD. Further research is warranted to isolate and characterize the active compounds responsible for this inhibitory activity and to evaluate their efficacy and safety in preclinical studies.

Our research offers a theoretical framework for the creation and application of *P. crotalarioides*’ endophytic fungal resources. The following areas of investigation could be pursued in the future: (1) further isolation and identification of culturable endophytic fungi to find prospective bioresources; (2) correlation of endophytic fungal community and diversity to the growth and active component accumulation of *P. crotalarioides*; and (3) discovery of bioactive compounds with anti-AChE activity in screened endophytic fungi.

## 5. Conclusions

To conclude, using HTS, a total of 1190 OTUs were annotated from *P. crotalarioides* samples at two growth stages of 1 and 2 years old, belonging to six phyla, 23 classes, 72 orders, 160 families, 290 genera, and 416 species with the unclassified taxa excluded. The dominant phylum of the endophytic fungi in *P. crotalarioides* was Ascomycota. In either 1- or 2-year-old samples, the three organs (leaf, stem, and root) of *P. crotalarioides* possessed different endophytic fungal combinations, with different dominant and endemic genera, demonstrating organ specificity or preference, which is an influential factor to explain the variation of the *P. crotalarioides* fungal community. In the same organs, the endophytic fungal communities varied with the plant growing ages. Furthermore, 55 fungi were isolated and identified from leaves, stems, and roots of *P. crotalarioides* using culture-based methods. The sequence analysis showed that all isolated endophytic fungi belong to the phyla Ascomycota and Basidiomycota. At the genus level, the most common endophytic fungi were *Fusarium* and *Aspergillus*, while five isolates exhibited AChE-inhibitory activity.

## Figures and Tables

**Figure 1 jof-10-00195-f001:**
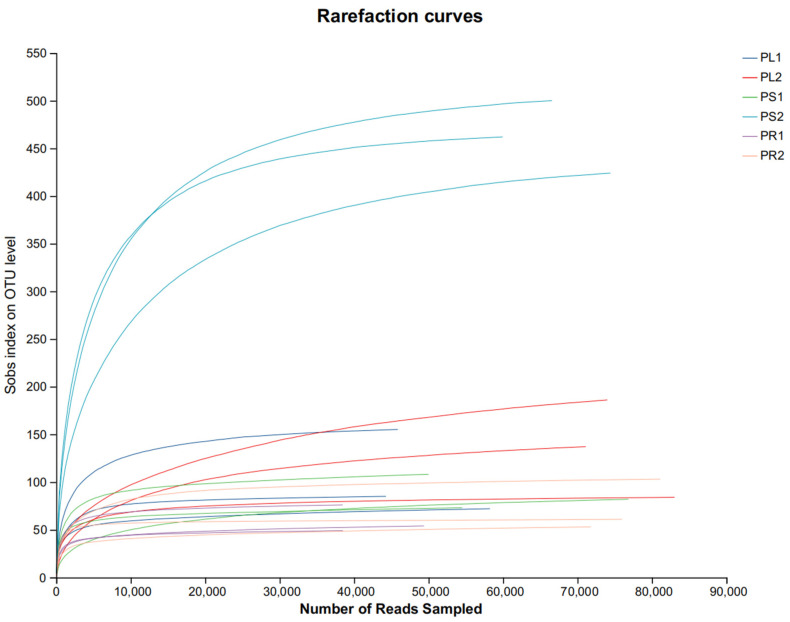
Rarefaction curves of endophytic fungi of 18 samples in *P. crotalarioides* based on sobs index. The abscissa represents the number of reads, and the ordinate represents the sobs index on OTU level. PL, PS, and PR stand for leaves, stems, and roots; the number after the abbreviations mean the sample growth stage is of 1 or 2 years.

**Figure 2 jof-10-00195-f002:**
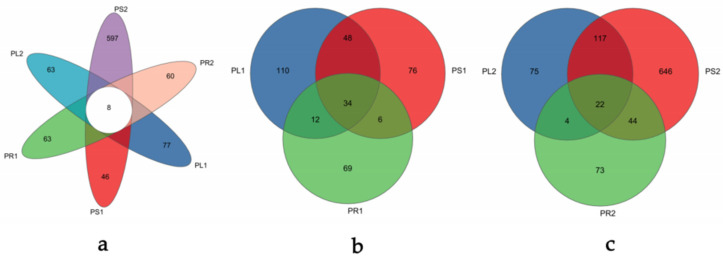
Venn diagram showing the endophytic fungal OTUs of all samples at 1- and 2-year-old stages (**a**), fungal OTUs of different organs in 1-year-old samples (**b**), and fungal OTUs of different organs in 2-year-old samples (**c**).

**Figure 3 jof-10-00195-f003:**
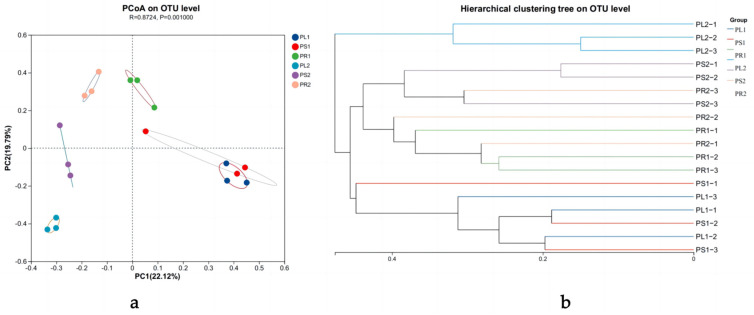
Principal coordinates analysis on OTU level (**a**). Both selected principal component axes are represented by the *x*−axis and *y*−axis, and percentage represents the difference in sample composition using principal components; scales of *x*−axis and *y*−axis represent relative distances. Samples are represented by different color points in different groups. Closeness of points represents similarity level between fungal species composition. Cluster analysis on OTU level (**b**).

**Figure 4 jof-10-00195-f004:**
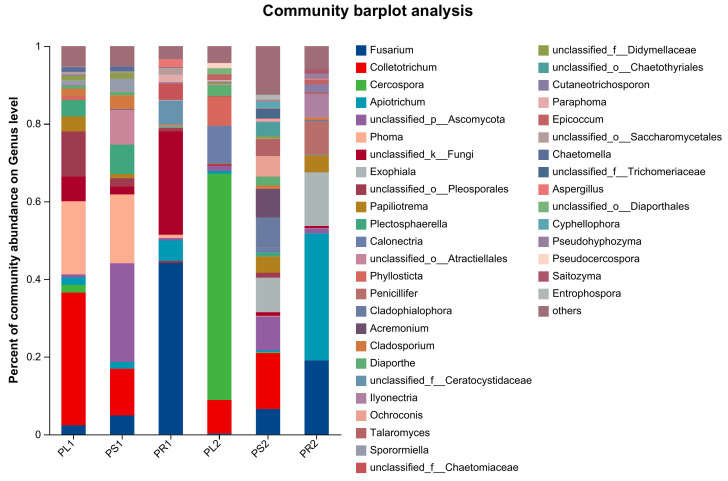
Relative abundances of the dominant fungal genus of different tissue samples. The abscissa represents sample name, and the ordinate represents the percentage of different species represented by columns with different colors, sizes, and proportions for a species.

**Table 1 jof-10-00195-t001:** Endophytic fungal community diversity and richness indices of *P. crotalarioides*.

Index	PL1	PS1	PR1	PL2	PS2	PR2
shannon	2.191 ± 0.63 ^b^	2.11 ± 0.91 ^b^	1.71 ± 0.61 ^b^	1.51 ± 0.71 ^b^	3.61 ± 0.31 ^a^	1.91 ± 0.51 ^b^
simpson	0.266 ± 0.16 ^a^	0.31 ± 0.21 ^a^	0.41 ± 0.21 ^a^	0.51 ± 0.31 ^a^	0.11 ± 0.01 ^a^	0.31 ± 0.21 ^a^
chao	108.7 ± 42.46 ^bc^	99.71 ± 15.61 ^bc^	62.11 ± 14.71 ^c^	153.41 ± 65.91 ^b^	474.61 ± 36.41 ^a^	79.71 ± 26.41 ^bc^
ace	112.7 ± 40.18 ^bc^	99.11 ± 16.41 ^bc^	63.51 ± 14.41 ^c^	154.01 ± 69.01 ^b^	473.91 ± 35.01 ^a^	83.71 ± 22.21 ^bc^
coverage	0.9998 ± 0.00004	0.9998 ± 0.00004	0.9999 ± 0.0001	0.9997 ± 0.00003	0.9999 ± 0.0001	0.9999 ± 0.00005

Lowercase letters indicate significant differences between samples (*p* < 0.05).

**Table 2 jof-10-00195-t002:** The main common genera of endophytic fungi among/between different samples.

Group Label	Number of Common Genera	Top 5 Genera and Their Relative Abundance
PL1:PS1:PR1	29	*Fusarium* (19.14%)
		*Colletotrichum* (17.31%)
		*Phoma* (13.91%)
		*Plectosphaerella* (4.47%)
		*Apiotrichum* (3.35%)
PL2:PS2:PR2	27	*Apiotrichum* (28.24%)
		*Fusarium* (21.56%)
		*Calonectria* (9.37%)
		*Papiliotrema* (7.37%)
		*Acremonium* (6.13%)
PL1:PL2	43	*Cercospora* (34.57%)
		*Colletotrichum* (24.54%)
		*Phoma* (10.81%)
		*Phyllosticta* (4.96%)
		*Diaporthe* (2.09%)
PS1:PS2	65	*Colletotrichum* (15.84%)
		*Phoma* (10.71%)
		*Fusarium* (6.82%)
		*Plectosphaerella* (5.13%)
		*Cladophialophora* (4.64%)
PR1:PR2	24	*Fusarium* (43.62%)
		*Apiotrichum* (26.24%)
		*Papiliotrema* (3.12%)
		*Aspergillus* (1.61%)
		*Cutaneotrichosporon* (1.60%)

**Table 3 jof-10-00195-t003:** Phylogenetic analysis and anti-AChE activity determination of culturable endophytic fungi from *P. crotalarioides*.

Source	Isolates	Closest Relatives in NCBI (Accession No.)	GenBank No.	Identity%	AChE Inhibition%
1-year-old leaf	PCAM003	*Fusarium tricinctum* (MK962346)	OR592306	100.00	NA
	PCAM010	*Phialophora mustea* (MK102700)	OR592307	99.83	54.02 ± 1.42
	PCAM033	*Didymella bellidis* (MN274965)	OR592308	100.00	NA
	PCAM034	*Phialophora* sp. (MT576439)	OR592309	100.00	22.87 ± 3.16
	PCAM073	*Pezicula chiangraiensis* (KU310621)	OR592310	100.00	14.10 ± 4.01
	PCAM074	*Fusarium tricinctum* (MK962346)	OR592311	100.00	NA
	PCAM078	*Didymella bellidis* (MN274965)	OR592312	100.00	NA
	PCAM087	*Talaromyces funiculosus* (MT367866)	OR592313	100.00	NA
	PCAM091	*Didymella bellidis* (MN274965)	OR592314	100.00	NA
	PCAM095	*Coniochaeta velutina* (MK656231)	OR592315	100.00	NA
1-year-old stem	PCAP001	*Phialophora mustea* (MK102700)	OR592316	100.00	NA
	PCAP003	*Fusarium reticulatum* (MT601889)	OR592317	99.81	11.05 ± 0.40
	PCAP012	*Geotrichum candidum* (KT921188)	OR592318	100.00	NA
	PCAP033	*Fusarium flocciferum* (MG386078)	OR592319	100.00	NA
	PCAP036	*Juxtiphoma* sp. (MK100167)	OR592320	100.00	23.28 ± 0.99
	PCAP041	*Alternaria alternata* (OQ248210)	OR592321	100.00	19.37 ± 0.51
	PCAP093	*Pezicula sporulosa* (MW487233)	OR592322	99.06	NA
	PCAP096	*Fusarium reticulatum* (MT601889)	OR592323	99.81	14.54 ± 1.95
	PCAR020	*Coniochaeta velutina* (MK656231)	OR592324	100.00	NA
1-year-old root	PCBM001	*Botrytis cinerea* (KR002909)	OR592325	99.62	NA
	PCBM022	*Colletotrichum dematium* (MT446146)	OR592326	100.00	NA
	PCBM027	*Diaporthe nobilis* (KX866924)	OR592327	98.91	65.92 ± 2.29
	PCBM035	*Botrytis cinerea* (MT573470)	OR592328	100.00	NA
	PCBP027	*Talaromyces muroii* (KU744629)	OR592329	100.00	NA
	PCBR013	*Colletotrichum acutatum* (MN856423)	OR592330	100.00	NA
	PCBR020	*Cladosporium* sp. (OQ248224)	OR592331	100.00	NA
2-year-old leaf	LM17	*Alternaria alternata* (OW986459)	OR597909	100.00	24.03 ± 0.78
	LM35	*Leptosphaerulina arachidicola* (MK555325)	OR597910	100.00	NA
	LM80	Alternaria blumeae (MN612548)	OR597911	100.00	NA
	LP01	*Penicillium brefeldianum* (MH864250)	OR597912	100.00	NA
	LP24	*Colletotrichum horii* (MT568591)	OR597913	100.00	11.97 ± 2.19
	LP34	*Nigrospora chinensis* (MK371770)	OR597914	100.00	NA
	LP41	*Fusarium oxysporum* (MT530243)	OR597915	100.00	59.13 ± 0.83
	LP851	*Penicillium brefeldianum* MH864250	OR597916	100.00	NA
	LP87	*Pyronema omphalodes* (MK886722)	OR597917	100.00	NA
	LR101	*Nigrospora rubi* (NR_153470)	OR597918	100.00	44.73 ± 1.93
	LR43	*Nigrospora chinensis* (MK371770)	OR597919	100.00	NA
	LR51	*Colletotrichum gloeosporioides* (MT568599)	OR597920	100.00	13.96 ± 0.68
	LR62	*Nigrospora oryzae* (MW411591)	OR597921	100.00	NA
	LR87	*Colletotrichum boninense* (MT568597)	OR597922	100.00	NA
2-year-old stem	SM04	*Schizophyllum commune* (MK647986)	OR597926	100.00	NA
	SM21	*Nigrospora chinensis* (MN341444)	OR597927	100.00	NA
	SM40	*Nigrospora rubi* (MN486553)	OR597928	100.00	45.77 ± 0.82
	SM79	*Pestalotiopsis kenyana* (MN341553)	OR597929	100.00	NA
	SM81	*Phoma herbarum* (MT420621)	OR597930	100.00	89.88 ± 2.79
	SR58	*Chaetomium cochliodes* (MT520580)	OR597931	100.00	NA
	SR60	*Fusarium oxysporum* (MT530243)	OR597932	100.00	57.88 ± 1.85
2-year-old root	RM06	*Schizophyllum commune* (MT601949)	OR597923	100.00	NA
	RM07	*Epicoccum nigrum* (MN947593)	OR597924	100.00	NA
	RR09	*Aspergillus* sp. (LN898690)	OR597925	100.00	NA
	FM25	*Colletotrichum horii* (MT568591)	OR597904	100.00	10.80 ± 0.92
	FM83	*Phyllosticta sorghina* (MK762588)	OR597905	100.00	NA
	FP76	*Diaporthe unshiuensis* (MN816431)	OR597906	100.00	34.25 ± 2.09
	FR89	*Alternaria alternata* (OW986459)	OR597907	100.00	14.57 ± 1.22
	FR92	*Periconia pseudobyssoides* (MN944517)	OR597908	100.00	NA

The inhibition effects of fungal fermentation liquid on AChE activity in vitro were based on triplicate tests, values are mean ± SD of three replications. The inhibition effect of galantamine and PDB media was 96.25% ± 1.10% and <5%, respectively. NA—no significant activity; AChE inhibition rate < 10%.

## Data Availability

The sequence data from the endophytic fungi of *Polygala crotalarioides* were deposited in the Sequence Read Archive of the NCBI under accession number PRJNA973057. The ITS rRNA gene sequences of culturable endophytic fungal strains were deposited in the GenBank database with accession numbers OR592306-OR592331 and OR597904-OR597932.

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
