# Peer review of "Community Structure and Diversity of Endophytic Fungi in Cultivated Polygala crotalarioides at Two Different Growth Stages Based on Culture-Independent and Culture-Based Methods"

_jof, 2024, doi:10.3390/jof10030195_

Round 1

Reviewer 1 Report

This manuscript reports on the fungal endophytic associates of Polygala crotalarioides, as examined by both direct culturing and a metagenomic approach. Original findings on the composition of this component of the plant microbiota are reported with reference to the plant organ-age; besides the descriptive aims, the possible impact of the exploration of endophytic diversity is presented in terms of detection of AChE inhibitors, deserving consideration for possible publication in JoF. In this respect, the below adjustments are required for improving language style, fluency and conciseness.

Abstract

Lines 22-23 correct to ‘…using both culture-independent (high-throughput sequencing, HTS) and culture-based methods.’

Line 28: names of fungal genera require italics. This is to be checked throughout the manuscript (e.g. lines 102, 376, etc.).

Lines 30-32: this sentence should be displaced at line 26 (after ‘2-year plants.’) for a more logic sequence.

Introduction

Line 47-48: delete ‘. It is also estimated that four to five endophytic fungi live in each plant on average’, since the authors themselves demonstrate that this number can be much higher.

Lines 49-51: delete this sentence, since it has redundancy with previous and subsequent statements.

Lines 57-59: this sentence must be changed to a more correct form, such as ‘However, the traditional culturing methods limit the detection of endophytic fungi because isolation of many fungi is problematic’.

Line 60: delete ‘starting in the latter part of the 2000s’.

Line 65: delete ‘those of’.

Line 69: correct to ‘Arctic’.

Line 71: correct to ‘belonging’.

Lines 72-73: correct to '… China; among them Polygala crotalarioides Buch.-Ham. ex DC. is

widely distributed…’. The botanical family (Polygalaceae) should be better specified on the first mention of Polygala (line 66).

Line 75: correct to ‘Likewise other Polygala species, including P. tenuifolia [19] and P. japonica’.

Lines 78-80: delete these sentences.

Lines 86-87: correct to ‘More and more research found that Endophytic fungi are becoming a significant source of AChE inhibitors [25], and it is…’.

M&M

Line 170: correct to ‘purification’.

Results

Line 198: specify that you refer to the control of washing water.

Line 229: 7 phyla, 28 classes? I am perplexed since the use of ITS should only enable to identify fungi. However, supplementary tables are missing and I cannot perform the necessary check. This information must be checked again at line 287 and line 529

Line 233: correct to ‘the three organs’.

Lines 269-270: correct to ‘endophytic fungi were also observed in different organ groups except PL1 and PL2, regardless of whether…’.

Line 293: correct to ‘…organs changed with the stage of’.

Line 300: I find it correct to exclude unclassified, but consistency is needed with this choice; implying that it is INCORRECT to consider unclassified taxa of all levels in the following lines. Authors should limit to provide and discuss data concerning the identified genera; indeed, unidentified taxa are not genera, and must be excluded from the counting (line 317 and on).

Page 10: I remark that the final part of section 3.2 must be removed. In fact, identification is limited to the genus level, and guilds cannot be referred to genera! Actually, I find it quite intuitive that within fungal genera there is a wide variation in the ecological role and the functional relationships with plants, which make it very approximate, if not useless, to try ascription to any specific guild.

Table 3: valid names of some species must be checked in Mycobank. Particularly, correct name for Fusarium petersiae is Fusarium flocciferum, and correct name for Epicoccum sorghinum is Phyllosticta sorghina.

Section 3.4: in my opinion this section should be removed, since it is intuitive and logic that HTS allows the detection and identification of a much higher number of species.

Discussion

Line 421: delete ‘have’.

Line 422: correct ‘have produced’ to ‘may produce’.

Line 426: correct ‘underscored’ to ‘underlined’.

Lines 429-430: correct to ‘A typical example is huperzine A produced by Huperzia serrata, an AChE inhibitor used to treat…’.

Line 439: correct to ‘…elongata was reported as a potential…’.

Lines 447-450: delete this sentence, for the reasons anticipated in comment to section 3.4.

Line 454: correct to ‘Ascomycota’.

Lines 476-489: this text should be removed for the reasons anticipated in comment referring to page 10. This also applies to sentence at lines 498-499, point (3) at lines 524-525, and sentence at lines 536-537.

Line 500: correct to ‘…that composition of the endophytic fungal communities depends on the plant age’.

Line 504: correct 'existed’ to ‘displayed’.

Conclusions

Line 538: correct to ‘roots of’.

Line 539: correct ‘were attached’ to 'belong’.

Lines 540-541: correct to 'At the genus level, the most common endophytic fungi were Fusarium and Aspergillus, while five isolates exhibited AChE-inhibitory activity’.

Reviewer 2 Report

The article “Community Structure and Diversity of Endophytic Fungi in Cultivated Polygala crotalarioides at Two Different Growth Stages Based on Culture-Independent and Culture-Based Methods” by Kaize Shen, Yu Xiong, Yanfang Liu, Xingwang Fan, Rui Zhu, Zumao Hu, Congying Li and Yan Hua is devoted to theoretical and practical aspects of the structural composition, biodiversity and bioactivity of endophytic fungi in P. crotalarioides. The manuscript is well-structured and well-written and may be useful for medicine.

Specific points:

Lines 51-53. “During plant growth, endophytic fungi display many substantial benefits, such as accelerating plant growth, enhancing stress resistance, promoting nutrient absorption, and resisting pathogens [6].”

Endophytic fungi are extensively and actively researched worldwide. One reference is not sufficient. Please add references for each of the listings in the sentence:

accelerating plant growth

enhancing stress resistance

promoting nutrient absorption

resisting pathogens

Lines 75-76. “Like other plants belonged to the genus, including Polygala tenuifolia [19], Polygala japonica [20], P. crotalarioides has potential therapeutic effects on neuroprotection.”

Please insert a reference at the end of the sentence.

Lines 78-80. “Medicinal plants have become an important resource for discovering endophytic fungi with bioactivity.”

Please insert a reference at the end of the sentence.

Lines 86-87. “More and more research found that endophytic fungi are becoming a significant source of AChE inhibitors [25].”

If you write “more and more” it means that one reference is not enough. Please insert references at the end of the sentence.

Lines 89-90. "Normally, P. crotalarioides must develop for at least two years before it may be utilized medically….”

Please briefly state the reason why plants are not used in medicine before the age of two. Add a reference.

Line 105.  “….within 24 h under low temperature.”

Please enter a specific temperature value

Lines 154-155. Please change petri dishes to Petri dishes

Line 202. A total of 18 samples were analysed. Please break down how many leaf samples, how many root samples, how many stem samples.

Line 213-214. Please indicate the abbreviations PL PS PS PR in the legend to Figure one.

Lines 316-317. “At genus level, PL1 had 99 genera that was much more than PL2 with 92 genera”

I am not sure whether it is right to write “much more” in this case.

Lines 327-328. In Table 2, in column “Top 5 genera and their relative abundance” please put the names of the genera one below the other in a column.

Lines 339-350. It is not necessary to duplicate the information from Figure 5 (list of functional groups) in the text. A reference to the figure is sufficient.

Lines 397-398. “Overall, culture-independent method detected much more species compared to culture-based method.”

This does not seem to me to be a valid conclusion. With one method you only get a list of OTUs, no specific species, with the other specific species. The methods complement each other and add to the understanding and not one is better than the other. With the isolated fungal strains, specific studies of medical and biotechnological importance can be carried out. OTU diversity provides general insights into genetic and ecological diversity.

Line 454. Please correct “Ascomata”.

Line 524. Please italic the Latin name “accumulation of P. crotalarioides”.

The article “Community Structure and Diversity of Endophytic Fungi in Cultivated Polygala crotalarioides at Two Different Growth Stages Based on Culture-Independent and Culture-Based Methods” by Kaize Shen, Yu Xiong, Yanfang Liu, Xingwang Fan, Rui Zhu, Zumao Hu, Congying Li and Yan Hua is devoted to theoretical and practical aspects of the structural composition, biodiversity and bioactivity of endophytic fungi in P. crotalarioides. The manuscript is well-structured and well-written and may be useful for medicine.

Specific points:

Lines 51-53. “During plant growth, endophytic fungi display many substantial benefits, such as accelerating plant growth, enhancing stress resistance, promoting nutrient absorption, and resisting pathogens [6].”

Endophytic fungi are extensively and actively researched worldwide. One reference is not sufficient. Please add references for each of the listings in the sentence:

accelerating plant growth

enhancing stress resistance

promoting nutrient absorption

resisting pathogens

Lines 75-76. “Like other plants belonged to the genus, including Polygala tenuifolia [19], Polygala japonica [20], P. crotalarioides has potential therapeutic effects on neuroprotection.”

Please insert a reference at the end of the sentence.

Lines 78-80. “Medicinal plants have become an important resource for discovering endophytic fungi with bioactivity.”

Please insert a reference at the end of the sentence.

Lines 86-87. “More and more research found that endophytic fungi are becoming a significant source of AChE inhibitors [25].”

If you write “more and more” it means that one reference is not enough. Please insert references at the end of the sentence.

Lines 89-90. "Normally, P. crotalarioides must develop for at least two years before it may be utilized medically….”

Please briefly state the reason why plants are not used in medicine before the age of two. Add a reference.

Line 105.  “….within 24 h under low temperature.”

Please enter a specific temperature value

Lines 154-155. Please change petri dishes to Petri dishes

Line 202. A total of 18 samples were analysed. Please break down how many leaf samples, how many root samples, how many stem samples.

Line 213-214. Please indicate the abbreviations PL PS PS PR in the legend to Figure one.

Lines 316-317. “At genus level, PL1 had 99 genera that was much more than PL2 with 92 genera”

I am not sure whether it is right to write “much more” in this case.

Lines 327-328. In Table 2, in column “Top 5 genera and their relative abundance” please put the names of the genera one below the other in a column.

Lines 339-350. It is not necessary to duplicate the information from Figure 5 (list of functional groups) in the text. A reference to the figure is sufficient.

Lines 397-398. “Overall, culture-independent method detected much more species compared to culture-based method.”

This does not seem to me to be a valid conclusion. With one method you only get a list of OTUs, no specific species, with the other specific species. The methods complement each other and add to the understanding and not one is better than the other. With the isolated fungal strains, specific studies of medical and biotechnological importance can be carried out. OTU diversity provides general insights into genetic and ecological diversity.

Line 454. Please correct “Ascomata”.

Line 524. Please italic the Latin name “accumulation of P. crotalarioides”.

Round 2

Reviewer 1 Report

I verified that authors incorporated most of the required adjustments in their modified version, which is now acceptable for publication. They only need to check again the numbers of phyla, classes and other taxa in statements at lines 232, 291 and 539, considering that the 'unclassified' ones must not be included in the counting. Moreover, I insist that the present tense 'belong' is to be used at line 549.

See major comments.